⊘ | **Open Peer Review** | Microbial Pathogenesis | Research Article

# Nutrient acquisition drives *Edwardsiella tarda* pathogenesis in necrotizing soft tissue infection

Kohei Yamazaki,[1] Takuya Yamaguchi,[1] Yuichi Yokoyama,[1] Yuka Tonosaki,[1] Klara Kursanbaeva,[1,2] Daisuke Motooka,[3] Yukihiro Akeda,[4,5,6] Takashige Kashimoto[1]

**ABSTRACT** Necrotizing soft tissue infections (NSTIs) are rapidly progressive and life-threatening diseases caused by diverse bacterial pathogens. While classical virulence factors, such as toxins and secretion systems, have been extensively characterized, the role of metabolic fitness in supporting bacterial survival within the nutrient-restricted host environment remains underexplored. *Edwardsiella tarda*, a human-pathogenic bacterium implicated in NSTIs, represents an emerging model for studying non-canonical pathogenic strategies. Here, we employed transposon-directed insertion site sequencing (TraDIS) to identify genes critical for *E. tarda* survival in a murine soft tissue infection model. A genome-wide screen revealed 41 genes significantly depleted during the infection, including those involved in iron and zinc acquisition (*fetB*, *zupT*), vitamin biosynthesis (*pdxK*, *cobA*), and polyamine metabolism (*speB*). Functional assays using defined minimal media demonstrated that supplementation with vitamin B6 or putrescine enhanced bacterial growth, validating their contribution to fitness under nutrient-limited conditions. Our findings indicate that *E. tarda* pathogenesis is driven not solely by classical virulence factors but also by its ability to acquire essential nutrients and adapt metabolically to host-imposed nutritional stress. This study provides the first genome-wide fitness map for *E. tarda* during soft tissue infection and reveals new targets for therapeutic intervention that disrupt nutrient acquisition systems. These results also emphasize the broader relevance of metabolic adaptation as a determinant of virulence in invasive bacterial infections.

**IMPORTANCE** Necrotizing soft tissue infections (NSTIs) are severe, rapidly progressing bacterial infections with high morbidity and mortality. Although classical virulence factors such as toxins have been widely studied, much less is known about how pathogens adapt metabolically to survive within the nutrient-restricted environment in host tissues. This study uses *Edwardsiella tarda*, an emerging NSTI pathogen, as a model to identify genes required for *in vivo* fitness using transposon insertion sequencing. By revealing the critical roles of nutrient acquisition and metabolic adaptation, rather than toxin production alone, this work challenges conventional paradigms of bacterial virulence. Our findings suggest that targeting bacterial nutrient acquisition pathways may offer a novel therapeutic approach to control invasive infections. Furthermore, this study provides the first genome-wide fitness map of *E. tarda* during soft tissue infection, offering a valuable resource for future research into polymicrobial wound infections and host–pathogen nutrient competition.

**KEYWORDS** soft tissue infection, transposons, signature-tagged mutagenesis

N ecrotizing soft tissue infections (NSTIs) are rapidly progressing, life-threatening bacterial infections characterized by extensive tissue necrosis and systemic toxicity (1, 2). While classical virulence factors, such as toxins and secretion systems, have been

**Peer Reviewer** Keita Nishiyama, Tohoku Daigaku, Aobaku, Sendai, Japan

Address correspondence to Takashige Kashimoto, kashimot@vmas.kitasato-u.ac.jp.

The authors declare no conflict of interest.

See the funding table on p. 16.

extensively studied in NSTI pathogens, such as group A streptococci, *Staphylococcus aureus*, *Vibrio vulnificus*, and *Aeromonas hydrophila* (3–7), recent evidence suggests that metabolic fitness and nutrient acquisition systems are equally essential for bacterial survival and proliferation within host tissues (8, 9). The host environment imposes multiple nutritional and oxidative stresses that restrict bacterial growth (10, 11). In particular, the host tightly limits access to essential nutrients such as iron, zinc, polyamines, and B vitamins as part of nutritional immunity (12, 13). This situation is further complicated in polymicrobial infections, which are frequently observed in wound-associated NSTIs, where interspecies nutrient competition is inevitable (14). To overcome these limitations, bacteria must deploy specialized transporters, biosynthetic pathways, and regulatory systems to adapt their physiology to the host milieu. However, the precise genetic determinants supporting bacterial persistence in soft tissue remain unclear.

While *Edwardsiella tarda* was historically classified as a fish pathogen, it has been increasingly recognized as an emerging zoonotic agent causing severe infections in humans (15–24). Recent taxonomic revisions have reclassified fish-pathogenic strains under the new genera *E. piscicida*, *E. anguillarum*, and *E. ictaluri*, while strains capable of infecting humans retain the name *E. tarda* (25). Clinical manifestations include gastroenteritis, bacteremia, hepatobiliary infections, and life-threatening soft tissue infections such as necrotizing fasciitis and myonecrosis (15–17, 19–22). Reported cases span various geographic regions, including Japan, the United States, and China (17, 18, 20, 24). A retrospective study from Japan identified 26 cases of *E. tarda* bacteremia over 12 years (17), with high mortality rates, particularly among patients with liver disease or immunosuppression (20, 21). Infections are often associated with exposure to aquatic environments or consumption of raw seafood (18, 23). Despite the clinical significance of these cases, the genetic and metabolic determinants that enable *E. tarda* to survive and proliferate in soft tissue remain poorly defined.

In this study, we applied transposon-directed insertion site sequencing (TraDIS) to comprehensively identify *E. tarda* genes required for survival in the soft tissue environment (26, 27). By comparing mutant frequencies between *in vitro* cultures and bacteria recovered from murine muscle tissue, we systematically screened for genes critical for *in vivo* fitness. Functional categorization of the attenuated mutants revealed an enrichment of genes involved in amino acid and polyamine metabolism, vitamin biosynthesis, metal ion transport, and transcriptional regulation.

Furthermore, selected candidate genes were evaluated through nutrient depletion assays using M9 minimal medium, allowing validation of their contribution to growth under defined nutrient-limited conditions. Collectively, our findings shed light on non-canonical fitness factors essential for bacterial colonization and proliferation in nutrient-restricted soft tissue environments.

## RESULTS

### Establishment of a murine NSTI model using *E. tarda*

*E. tarda* has been implicated as a causative agent of NSTIs. However, murine models that faithfully recapitulate the clinical features of NSTI remain scarce. To establish a reliable *in vivo* model, we evaluated the impact of bacterial pre-culture conditions on pathogenicity. To compare bacterial growth and motility under different nutritional conditions, we employed tryptic soy broth (TSB), a nutrient-rich medium containing peptides, amino acids, and glucose, and lactose broth (LaB), a nutritionally more limited medium in which lactose is the main carbon source. *E. tarda* is commonly cultivated in TSB (28), and growth was most robust in TSB at 37°C (data not shown). However, microscopic observation revealed that *E. tarda* cultured in LB at 25°C exhibited the highest motility (data not shown). Mice were subcutaneously inoculated with up to $10^7$ CFU/100 µL of *E. tarda* cultured under TSB or LaB. Bacteria grown in TSB at 37°C caused no lethality (Fig. 1A). Cultures grown in TSB at 25°C induced partial mortality, with several mice surviving. In contrast, cultures prepared in lactose broth (LaB) at 37°C resulted in >50% mortality within 24 h. Notably, LaB cultures at 25°C caused rapid and complete

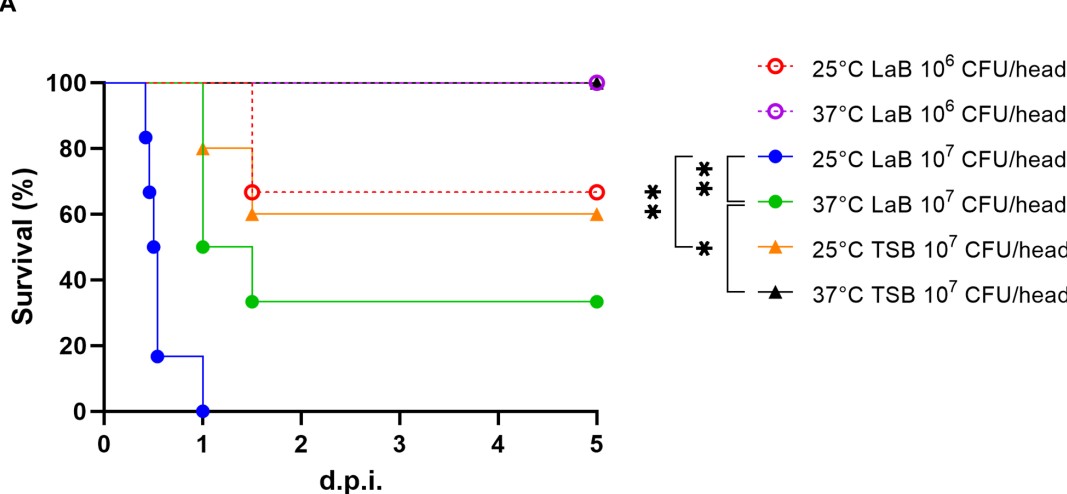

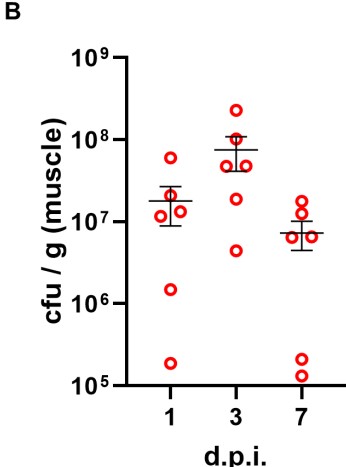

**FIG 1** Establishment of a murine model for *E. tarda*-induced NSTI. (A) Survival curves of mice subcutaneously infected with *E. tarda* pre-cultured in either TSB or LaB at 25°C or 37°C. Statistical significance was assessed using the log-rank test. (B) Bacterial burden (CFU per tissue) in infected muscle 1, 3, and 5 days post-infection. Bars represent means ± SEM from biological replicates. *$P < 0.05$, **$P < 0.01$.

lethality in all mice, indicating a highly virulent phenotype under these conditions (Fig. 1A). Histopathological analysis of infected tissues confirmed extensive muscle necrosis, validating the development of an NSTI-like condition (Fig. S1). Based on these findings, we selected *E. tarda* cultured in LaB at 25°C as the optimal pre-infection condition for modeling NSTI pathology *in vivo*.

Since the objective of the subsequent TraDIS was to identify bacterial genes required for *in vivo* growth, we aimed to establish an infection condition that would distinguish between attenuated and fit transposon mutants, a condition that is lethal yet allows sufficient bacterial proliferation in host tissues. In the inoculation of $10^6$ CFU of *E. tarda* cultured in LaB at 25°C, bacterial burden in infected muscle tissue increased until day 3 post-infection before declining (Fig. 1B), indicating that day 3 represents the peak of *in vivo* bacterial expansion. This condition induced consistent lethality while allowing sufficient dynamic range to assess mutant survival. Accordingly, soft tissue was harvested at 3 days post-infection as the optimal time point for TraDIS analysis.

## Genome-wide identification of fitness genes required for survival in soft tissue

To identify bacterial genes that are essential for pathogenesis in soft tissue, a transposon mutant library was constructed. The occurrence of random insertion of transposons across the genome was confirmed through the implementation of a Southern blotting (Fig. S2). To identify the genes required for *E. tarda* proliferation in soft tissue, TraDIS was performed. To circumvent potential bottlenecks, approximately 25,000 mutant clones were meticulously divided into three distinct groups and injected into mice. Three days post-infection, during the active growth phase *in vivo* (Fig. 1B), bacteria were recovered from muscle tissue. A negative selection analysis was then performed by comparing mutant frequencies between the input and output libraries. The TraDIS analysis with the input pool revealed insertions in 3,348 out of 3,473 predicted open reading frames (ORFs), indicating extensive genome coverage. Genes exhibiting a log2 fold change less than –1 and a $\log_{10}$ *P*-value greater than 1.3 were deemed to be significantly depleted *in vivo*. Among the 3,348 ORFs encompassed by the mutant library, a mere 41 genes (representing 1.22%) satisfied the stringent criteria of $\log_2$ fold change $\leq$ –1 and *P*-value < 0.05 (Fig. 2A), thereby signifying a highly selective set of genes deemed essential for *in vivo* fitness. These included genes involved in outer membrane biogenesis, particularly LPS-associated proteins that are critical for cell envelope stability and resistance to host antimicrobial factors (Fig. 2B; Table 1); nutrient acquisition systems, such as iron and zinc transporters (Fig. 2C; Table 2); vitamin-related metabolism including pyridoxal kinase and cob(I)alamin adenosyltransferase (Fig. 2D; Table 3); and stress response or transcriptional regulators that likely coordinate metabolic adaptation during infection (Fig. 2E; Table 4). These findings indicate that *E. tarda* requires only a restricted set of genes to adapt and proliferate within the soft tissue environment.

## Role of vitamin and polyamine metabolism in bacterial adaptation

TraDIS also identified several metabolic genes as critical for *in vivo* survival (Fig. 2D). The following genes were identified: pyridoxal kinase *pdxK* (vitamin B6 biosynthesis), *Cob*(I)alamin adenosyltransferase (vitamin B12 activation), and agmatinase *speB* (putrescine synthesis) (Fig. 2D; Table 3). In addition, genes linked to methionine and sulfur metabolism were identified, including 4-hydroxythreonine-4-phosphate dehydrogenase, thiosulfate sulfurtransferase GlpE, and anaerobic sulfite reductase subunit A. In order to ascertain the role of these nutrients in promoting growth, they were added individually to TSB and M9 media. No significant enhancement in growth was observed in TSB (Fig. 3A). However, in the M9 medium, growth assays indicated that putrescine supplementation significantly increased CFU compared with the control, whereas vitamin B6, vitamin B12, and methionine did not (Fig. 3B), suggesting that *E. tarda* can employ these compounds for growth in nutrient-restricted environments. These findings demonstrate that *E. tarda* relies not only on classical virulence systems but also on specific nutrient acquisition and metabolic pathways to survive and proliferate in the host.

## Iron and zinc acquisition systems are critical for *in vivo* fitness

Several genes were related to metal ion acquisition, including the ferric iron ABC transporter permease and the zinc importer ZupT. Furthermore, an isochorismatase family protein, which may play a role in siderophore biosynthesis, was identified (Fig. 2E, 4A and B; Table 4). The impact of metal availability on bacterial growth was assessed by supplementing or depleting iron and zinc in both nutrient-rich (TSB) and minimal (M9) media. In TSB, supplementation with iron or zinc led to a modest increase in OD600 (Fig. 4C). However, colony-forming unit (CFU) counts exhibited no growth enhancement (Fig. 4D). In contrast, the addition of 2,2′-dipyridyl to chelate iron or TPEN to chelate zinc resulted in a significant inhibition of growth, and this effect was reversed upon re-addition of the corresponding metal (Fig. 4D). In M9 medium, neither iron

supplementation nor 2,2′-dipyridyl treatment produced a significant effect on viable counts (Fig. 4E), indicating that *E. tarda* growth under minimal conditions was largely unaffected by iron availability. In M9 medium, the addition of zinc did not demonstrate an effect on the growth parameters. However, the application of TPEN resulted in a significant reduction in growth, a phenomenon that was rescued by the re-addition of zinc. These results underscore the significance of stringent regulation of iron and zinc acquisition for *E. tarda* survival *in vivo*.

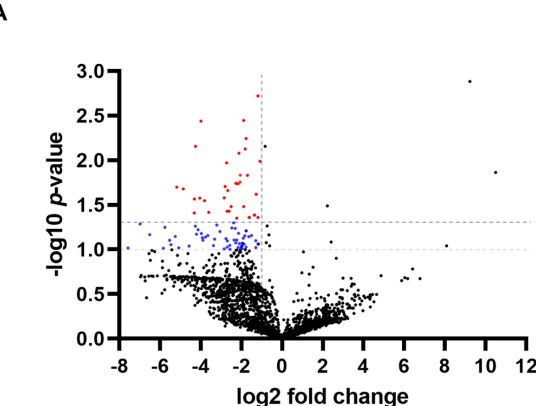

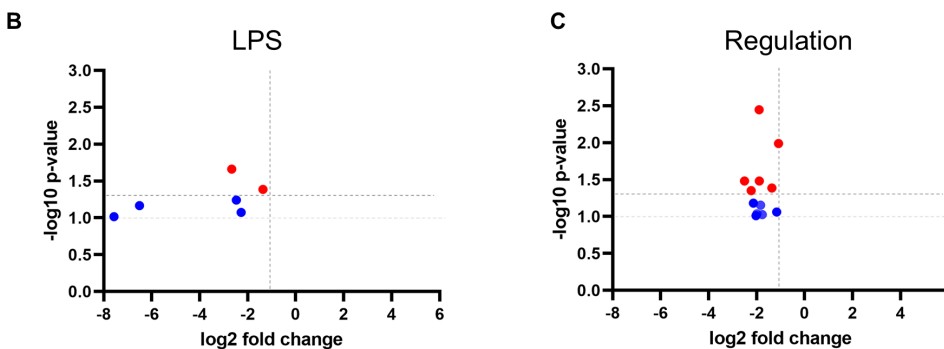

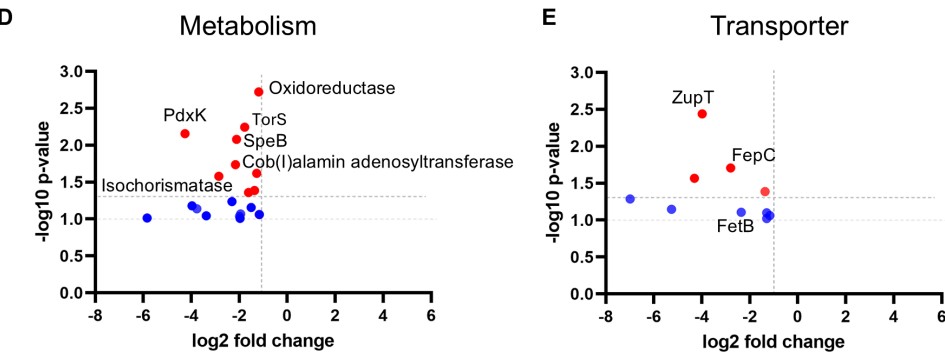

**FIG 2** Genome-wide identification of fitness genes required for *E. tarda* survival in soft tissue. (A) Volcano plot showing log$_2$ fold changes and −log$_{10}$ P-values for transposon insertion mutants recovered from *in vivo* versus *in vitro* conditions. Dotted lines indicate thresholds of log$_2$FC < −1 and −log$_{10}$ P > 1.3. Red dots indicate significantly depleted genes. Blue dots represent functionally related genes selected for further analysis (log$_2$FC < −1 and −log$_{10}$ P > 1.0). (B–E) Functional classification of selected genes: (B) outer membrane biosynthesis (LPS-related genes), (C) transcriptional regulators, (D) nutrient transporters, (E) metabolic enzymes.

**TABLE 1** Selected genes involved in outer membrane and LPS biosynthesis depleted *in vivo*

| Gene ID | Gene name | Log$_2$ fold change | *P*-value | Predicted pathway | KEGG pathway ID | Functional category | LPS/capsule detail |
|---|---|---|---|---|---|---|---|
| "RAST2:fig\|635.9.peg.2732" | UDP-3-O-[3-hydroxymyristoyl] glucosamine N-acyltransferase (EC 2.3.1.191) | −2.667 | 0.022 | Unknown | N/A$^a$ | LPS/Capsule | UDP-sugar Pathway |
| "RAST2:fig\|635.9.peg.50" | Lipopolysaccharide N-acetylmannosaminouronosyltransferase (EC 2.4.1.180) | −6.507 | 0.068 | Unknown | N/A | LPS/Capsule | Other LPS-related |
| "RAST2:fig\|635.9.peg.2643" | IncF plasmid conjugative transfer surface exclusion protein TraT | −2.277 | 0.084 | Unknown | N/A | LPS/Capsule | LPS Core/O-antigen Biosynthesis |
| "RAST2:fig\|635.9.peg.1386" | 3-Deoxy-manno-octulosonate cytidylyltransferase(EC 2.7.7.38) | −1.364 | 0.041 | Unknown | N/A | LPS/Capsule | Other LPS-related |
| "RAST2:fig\|635.9.peg.1637" | GtrA-like protein | −2.474 | 0.057 | Unknown | N/A | LPS/Capsule | LPS Core/O-antigen Biosynthesis |
| "RAST2:fig\|635.9.peg.2309" | UDP-4-amino-4-deoxy-L-arabinose formyltransferase (EC 2.1.2.13) / UDP-glucuronic acid oxidase (UDP-4-keto-hexauronic acid decarboxylating) (EC1.1.1.305) | −7.573 | 0.096 | Unknown | N/A | LPS/Capsule | UDP-sugar Pathway |

$^a$N/A, not applicable.

## *In vivo* attenuation of the Δ*zupT* mutant

To validate the role of zinc uptake in *E. tarda* pathogenesis, we constructed a *zupT* deletion mutant and assessed its virulence in the murine subcutaneous infection model. Survival analysis revealed that Δ*zupT* exhibited significantly reduced lethality compared with the wild-type (WT) strain (Fig. 5). These data provide direct *in vivo* confirmation that zinc acquisition contributes to bacterial fitness during necrotizing soft tissue infection.

**TABLE 2** Selected transcriptional regulators required for *E. tarda* survival in soft tissue

| Gene ID | Gene name | Log$_2$ fold change | *P*-value | Predicted pathway | KEGG pathway ID | Functional category | Regulatory detail |
|---|---|---|---|---|---|---|---|
| "RAST2:fig\|635.9.peg.2735" | Intramembrane protease RasP/YluC implicated in cell division based on FtsL cleavage | −2.226 | 0.044 | Unknown | N/A$^a$ | Regulatory Proteins | Other Regulation |
| "RAST2:fig\|635.9.peg.2909" | Protein lysine acetyltransferase Pat (EC2.3.1.-) | −2.138 | 0.066 | Unknown | N/A | Regulatory Proteins | Post-translational Modification |
| "RAST2:fig\|635.9.peg.2745" | [Protein-PII] uridylyltransferase (EC 2.7.7.59)/ [Protein-PII]-UMP uridylyl-removing enzyme | −1.836 | 0.07 | Unknown | N/A | Regulatory Proteins | Signal Transduction/Phosphorelay |
| "RAST2:fig\|635.9.peg.1133" | Phosphohistidine phosphatase SixA | −1.989 | 0.092 | Unknown | N/A | Regulatory Proteins | Signal Transduction/Phosphorelay |
| "RAST2:fig\|635.9.peg.1564" | Transcriptional regulator DeoR family | −1.894 | 0.004 | Unknown | N/A | Regulatory Proteins | Transcriptional Regulators |
| "RAST2:fig\|635.9.peg.2718" | Rho-specific inhibitor of transcription termination (YaeO) | −1.091 | 0.01 | Unknown | N/A | Regulatory Proteins | Transcriptional Regulators |
| "RAST2:fig\|635.9.peg.842" | Multidrug resistance regulator EmrR (MprA) | −1.893 | 0.033 | Unknown | N/A | Regulatory Proteins | Transcriptional Regulators |
| "RAST2:fig\|635.9.peg.1421" | Regulator of sigma S factor FliZ | −2.503 | 0.033 | Unknown | N/A | Regulatory Proteins | Transcriptional Regulators |
| "RAST2:fig\|635.9.peg.2493" | Phosphate regulon transcriptional regulatory protein PhoB (SphR) | −1.364 | 0.041 | Unknown | N/A | Regulatory Proteins | Transcriptional Regulators |
| "RAST2:fig\|635.9.peg.1485" | Transcriptional regulator | −1.169 | 0.087 | Unknown | N/A | Regulatory Proteins | Transcriptional Regulators |
| "RAST2:fig\|635.9.peg.1987" | Uncharacterized transcriptional regulator YozGCro/CI family | −1.766 | 0.094 | Unknown | N/A | Regulatory Proteins | Transcriptional Regulators |
| "RAST2:fig\|635.9.peg.2516" | Transcriptional regulator ArsR family | −2.031 | 0.098 | Unknown | N/A | Regulatory Proteins | Transcriptional Regulators |

$^a$N/A, not applicable.

**TABLE 3** Selected transporter genes involved in nutrient acquisition

| Gene ID | Gene name | Log2 fold change | P-value | Predicted pathway | KEGG pathway ID | Functional category | Functional subcategory | Metabolism detail |
|---|---|---|---|---|---|---|---|---|
| "RAST2:fig|635.9.peg.955" | Putative oxidoreductase SCO7655 | −1.189 | 0.002 | Unknown | N/Aª | Metabolism | Energy Metabolism | Energy Metabolism |
| "RAST2:fig|635.9.peg.3220" | Trimethylamine-N-oxide sensor histidine kinase TorS (EC 2.7.13.3) | −1.772 | 0.006 | Unknown | N/A | Metabolism | Nucleotide Metabolism | Other Metabolism |
| "RAST2:fig|635.9.peg.1808" | Pyridoxal kinase PdxK (EC 2.7.1.35) | −4.258 | 0.007 | Vitamin B6 metabolism | eco00750 | Metabolism | Vitamin/Coenzyme Metabolism | Vitamin/Coenzyme Metabolism |
| "RAST2:fig|635.9.peg.1583" | NADH:flavin oxidoreductases Old Yellow Enzyme family | −1.276 | 0.024 | Energy metabolism (Electron transport) | eco00190 | Metabolism | Energy Metabolism | Energy Metabolism |
| "RAST2:fig|635.9.peg.2119" | Cardiolipin synthase bacterial type ClsA | −2.304 | 0.058 | Glycerophospholipid metabolism | eco00564 | Metabolism | Other Metabolism | Other Metabolism |
| "RAST2:fig|635.9.peg.1687" | Dihydrodipicolinate synthase family proteinbII7272 | −1.505 | 0.07 | Lysine biosynthesis | eco00300 | Metabolism | Other Metabolism | Other Metabolism |
| "RAST2:fig|635.9.peg.388" | Phosphoenolpyruvate carboxykinase [ATP] (EC4.1.1.49) | −1.169 | 0.087 | Gluconeogenesis | eco00010 | Metabolism | Nucleotide Metabolism | Energy Metabolism |
| "RAST2:fig|635.9.peg.1127" | Cytochrome c-type biogenesis protein CcmG/DsbE thiol:disulfide oxidoreductase | −1.169 | 0.087 | Unknown | N/A | Metabolism | Energy Metabolism | Energy Metabolism |
| "RAST2:fig|635.9.peg.2738" | 1-deoxy-D-xylulose 5-phosphate reductoisomerase (EC 1.1.1.267) | −1.989 | 0.092 | Terpenoid backbone biosynthesis | eco00900 | Metabolism | Other Metabolism | Other Metabolism |
| "RAST2:fig|635.9.peg.2766" | Protease III precursor (EC 3.4.24.55) | −1.364 | 0.041 | Unknown | N/A | Metabolism | Amino Acid Metabolism | Other Metabolism |
| "RAST2:fig|635.9.peg.2782" | 2′–5′ RNA ligase | −1.364 | 0.041 | Unknown | N/A | Metabolism | Nucleotide Metabolism | Other Metabolism |
| "RAST2:fig|635.9.peg.1838" | NADP-dependent 3-hydroxy acid dehydrogenase YdfG (EC 1.1.1.381) 3-hydroxypropionate dehydrogenase (EC 1.1.1.298) | −1.61 | 0.044 | Unknown | N/A | Metabolism | Energy Metabolism | Energy Metabolism |
| "RAST2:fig|635.9.peg.646" | Agmatinase (EC 3.5.3.11) | −2.116 | 0.008 | Arginine and proline metabolism | eco00330 | Metabolism | Amino Acid Metabolism | Amino Acid Metabolism |
| "RAST2:fig|635.9.peg.2085" | Cob(I)alamin adenosyltransferase (EC 2.5.1.17) | −2.163 | 0.018 | Cobalamin (Vitamin B12) metabolism | eco00780 | Metabolism | Vitamin/Coenzyme Metabolism | Other Metabolism |
| "RAST2:fig|635.9.peg.1597" | Isochorismatase (EC 3.3.2.1) | −2.851 | 0.026 | Biosynthesis of siderophore group nonribosomal peptides | eco01053 | Metabolism | Siderophore/Iron Acquisition Metabolism | Sulfur/Iron Metabolism |
| "RAST2:fig|635.9.peg.2867" | 4-hydroxythreonine-4-phosphate dehydrogenase(EC 1.1.1.262) | −1.61 | 0.044 | Unknown | N/A | Metabolism | Vitamin/Coenzyme Metabolism | Energy Metabolism |
| "RAST2:fig|635.9.peg.2412" | Cytidine deaminase (EC 3.5.4.5) | −3.967 | 0.066 | Pyrimidine metabolism | eco00240 | Metabolism | Nucleotide Metabolism | Amino Acid Metabolism |
| "RAST2:fig|635.9.peg.357" | Thiosulfate sulfurtransferase GlpE (EC2.8.1.1) | −3.765 | 0.073 | Sulfur metabolism | eco00920 | Metabolism | Sulfur Metabolism | Sulfur/Iron Metabolism |
| "RAST2:fig|635.9.peg.409" | Phosphoglycolate phosphatase (EC 3.1.3.18) | −1.943 | 0.085 | Glyoxylate and dicarboxylate metabolism | eco00630 | Metabolism | Nucleotide/Intermediate Metabolism | Other Metabolism |
| "RAST2:fig|635.9.peg.2895" | Molybdopterin adenylyltransferase (EC2.7.7.75) | −3.379 | 0.091 | Molybdenum cofactor biosynthesis | eco00790 | Metabolism | Vitamin/Coenzyme Metabolism | Vitamin/Coenzyme Metabolism |
| "RAST2:fig|635.9.peg.2814" | Pyruvate dehydrogenase E1 component (EC1.2.4.1) | −5.837 | 0.097 | Unknown | N/A | Metabolism | Energy Metabolism | Energy Metabolism |

*(Continued on next page)*

**TABLE 3** Selected transporter genes involved in nutrient acquisition (*Continued*)

| Gene ID | Gene name | Log2 fold change | P-value | Predicted pathway | KEGG pathway ID | Functional category | Functional subcategory | Metabolism detail |
|---|---|---|---|---|---|---|---|---|
| "RAST2:fig\|635.9.peg.1849" | Anaerobic sulfite reductase subunit A | −1.963 | 0.098 | Unknown | N/A | Metabolism | Energy Metabolism | Sulfur/Iron Metabolism |
| "RAST2:fig\|635.9.peg.955" | Putative oxidoreductase SCO7655 | −1.189 | 0.002 | Unknown | N/A | Metabolism | N/A | Energy Metabolism |
| "RAST2:fig\|635.9.peg.3220" | Trimethylamine-N-oxide sensor histidine kinase TorS (EC 2.7.13.3) | −1.772 | 0.006 | Unknown | N/A | Metabolism | N/A | Nucleotide Metabolism |
| "RAST2:fig\|635.9.peg.1808" | Pyridoxal kinase pdxK (EC 2.7.1.35) | −4.258 | 0.007 | Vitamin B6 metabolism | eco00750 | Metabolism | N/A | Vitamin/Coenzyme Metabolism |
| "RAST2:fig\|635.9.peg.1583" | NADH:flavin oxidoreductases Old Yellow Enzyme family | −1.276 | 0.024 | Energy metabolism (Electron transport) | eco00190 | Metabolism | N/A | Energy Metabolism |
| "RAST2:fig\|635.9.peg.2119" | Cardiolipin synthase bacterial type ClsA | −2.304 | 0.058 | Glycerophospholipid metabolism | eco00564 | Metabolism | N/A | Other Metabolism |
| "RAST2:fig\|635.9.peg.1687" | Dihydrodipicolinate synthase family proteinbII7272 | −1.505 | 0.07 | Lysine biosynthesis | eco00300 | Metabolism | N/A | Other Metabolism |
| "RAST2:fig\|635.9.peg.388" | Phosphoenolpyruvate carboxykinase [ATP] (EC4.1.1.49) | −1.169 | 0.087 | Gluconeogenesis | eco00010 | Metabolism | N/A | Nucleotide Metabolism |
| "RAST2:fig\|635.9.peg.1127" | Cytochrome c-type biogenesis protein CcmG/DsbE thiol:disulfide oxidoreductase | −1.169 | 0.087 | Unknown | N/A | Metabolism | N/A | Energy Metabolism |
| "RAST2:fig\|635.9.peg.2738" | 1-deoxy-D-xylulose 5-phosphate reductoisomerase(EC 1.1.1.267) | −1.989 | 0.092 | Terpenoid backbone biosynthesis | eco00900 | Metabolism | N/A | Other Metabolism |

[a]N/A, not applicable.

**TABLE 4** Selected metabolic genes associated with vitamin, sulfur, and amino acid pathways

| Gene ID | Gene name | Log$_2$ fold change | P-value | Predicted pathway | KEGG pathway ID | Functional category | Functional subcategory | Transporter detail |
|---|---|---|---|---|---|---|---|---|
| "RAST2:fig\|635.9.peg.1502" | PTS system fructose-like IIC component FrwC | −1.364 | 0.041 | Phosphotransferase system (PTS) | eco02060 | Transporters | N/A[a] | Sugar/Carbohydrate Transport |
| "RAST2:fig\|635.9.peg.2991" | Zinc transporter ZupT | −3.986 | 0.004 | Unknown | N/A | Transporters | N/A | Iron/Zinc Transport |
| "RAST2:fig\|635.9.peg.838" | Ferric iron ABC transporter permease protein | −2.799 | 0.02 | ABC transporters | eco02010 | Transporters | Siderophore/Iron Acquisition metabolism | Iron/Zinc Transport |
| "RAST2:fig\|635.9.peg.1532" | Putative xanthosine permease | −4.309 | 0.027 | Unknown | N/A | Transporters | N/A | ABC Transporters |
| "RAST2:fig\|635.9.peg.1442" | LSU ribosomal protein L32p LSU ribosomal protein L32p zinc-independent | −5.261 | 0.071 | Unknown | N/A | Transporters | N/A | Iron/Zinc Transport |
| "RAST2:fig\|635.9.peg.1425" | Probable iron export permease protein FetB | −2.36 | 0.078 | Unknown | N/A | Transporters | N/A | Iron/Zinc Transport |
| "RAST2:fig\|635.9.peg.153" | Ribose ABC transporter ATP-binding protein RbsA (TC 3.A.1.2.1) | −1.296 | 0.08 | ABC transporters | eco02010 | Transporters | N/A | ABC Transporters |
| "RAST2:fig\|635.9.peg.1940" | Na(+)-linked D-alanine glycine permease | −1.169 | 0.087 | Unknown | N/A | Transporters | N/A | ABC Transporters |
| "RAST2:fig\|635.9.peg.2894" | L-Proline/Glycine betaine transporter ProP | −1.296 | 0.095 | Unknown | N/A | Transporters | N/A | Other Transporters |
| "RAST2:fig\|635.9.peg.11" | Twin-arginine translocation protein TatC | −6.987 | 0.052 | Unknown | N/A | Transporters | N/A | Other Transporters |

[a]N/A, not applicable.

## DISCUSSION

Our TraDIS screen identified multiple genes related to metal ion transport and homeostasis as essential for *E. tarda* survival during soft tissue infection. Notably, *fepC*, encoding a ferric iron ABC transporter permease, and *zupT*, a zinc importer, were among the most significantly depleted *in vivo* (Fig. 2A and E). These findings underscore the critical role of iron and zinc acquisition under host-imposed nutrient limitation, a phenomenon well-documented in deep tissue infections (10–13, 29, 30).

Before TraDIS screening, we established a reproducible murine infection model that accurately reflects the pathology of NSTIs (Fig. S1). Interestingly, we found that virulence was strongly influenced by bacterial pre-culture conditions (Fig. 1A). While *E. tarda* grown in TSB at 37°C exhibited negligible pathogenicity in mice, cultures prepared in LaB, particularly at 25°C, resulted in rapid and complete mortality (Fig. 1A). Thus, we observed a striking difference in mouse survival depending on the pre-infection culture medium (LaB and TSB) and temperature (25°C vs 37°C). Microscopic observation further revealed that *E. tarda* cultured in LaB at 25°C displayed the highest motility, whereas growth was most robust in TSB at 37°C. This contrast suggests that environmental priming may differentially regulate motility and proliferation before infection. Temperature shifts are known to affect bacterial physiology, including remodeling of surface structures and LPS composition (31, 32), induction of virulence-associated genes (33), and metabolic pre-adaptation (34). Importantly, our previous work demonstrated that motility accelerates the progression of necrosis in a murine wound infection model of *Aeromonas hydrophila* (7) and that chemotactic invasion is essential for deep tissue spread by *Vibrio vulnificus* (6). Together, these findings support the hypothesis that temperature-dependent enhancement of motility contributes to the virulence potential of *E. tarda* during NSTIs.

Several genes associated with LPS biosynthesis and outer membrane stability were significantly depleted *in vivo*. Since LPS is both a structural component and a major target of host immunity, these findings suggest that maintaining envelope integrity is critical for *E. tarda* survival in soft tissue. Similar conclusions have been drawn for *Vibrio vulnificus* and *Salmonella enterica*, where defects in LPS biosynthesis markedly increase susceptibility to complement and antimicrobial peptides (35, 36). We also identified transcriptional regulators and stress-response genes that were selectively depleted *in vivo*. These factors likely coordinate adaptive programs that allow *E. tarda* to remodel its metabolism and envelope composition under nutrient limitation and host-derived stresses. Comparable mechanisms have been described in *Salmonella enterica* and *Vibrio*

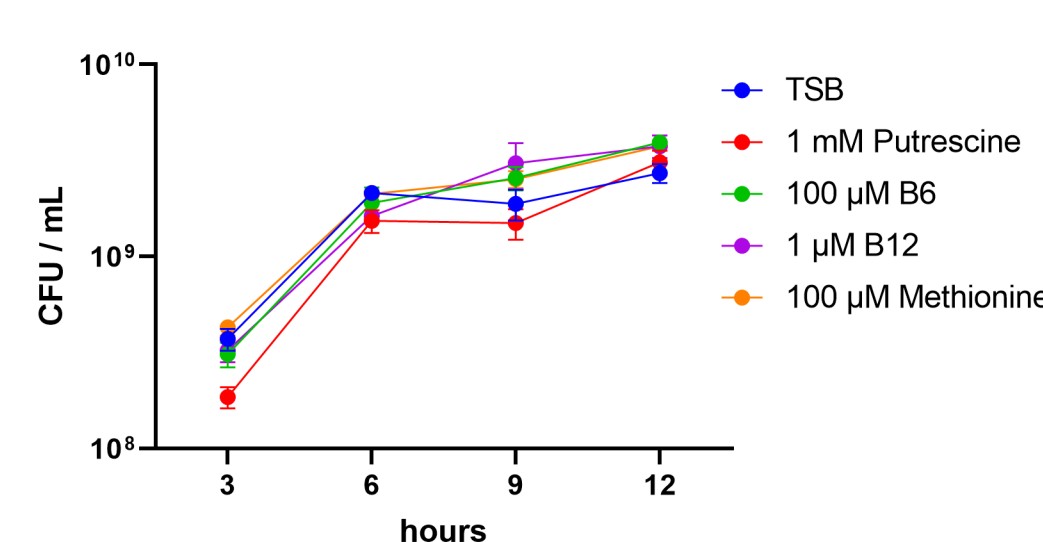

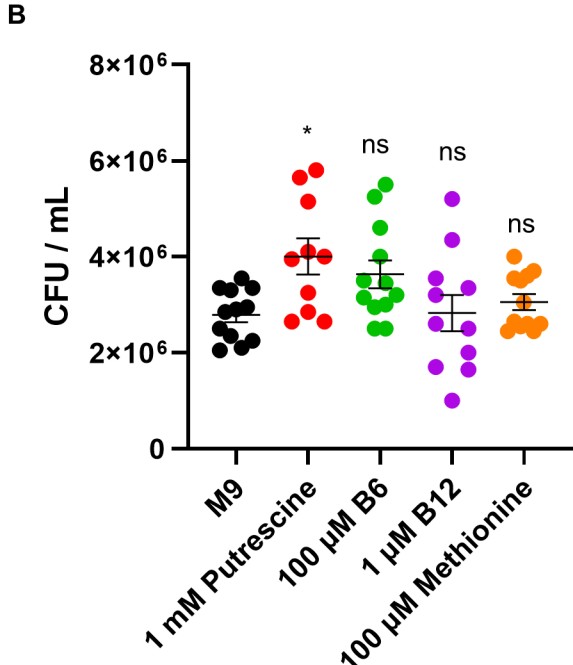

**FIG 3** Putrescine enhances the growth of *E. tarda* in minimal medium. CFU counts of *E. tarda* cultured in TSB (A) or M9 minimal medium (B) supplemented with nutrients. Bars represent means ± SEM from biological replicates. Statistical analysis was performed using the Kruskal–Wallis test followed by Dunn's multiple-comparison test with Holm–Šidák correction (each treatment vs control).*$P < 0.05$; ns, not significant.

*cholerae*, where transcriptional regulators integrate environmental cues with virulence gene expression (32, 37). These findings complement our analysis of nutrient acquisition pathways, described below.

Our screen identified key metabolic genes: *pdxK* (vitamin B6), *cobA* (vitamin B12), and *speB* (polyamine biosynthesis). These represent categories significantly depleted in the TraDIS screen, namely micronutrient metabolism and polyamine utilization. Supplementing M9 minimal medium with pyridoxal or putrescine partially restored growth (Fig. 3B; Table 3), confirming a functional requirement for these micronutrients. This supports previous findings in UPEC and *P. aeruginosa* that underscore the importance of vitamin

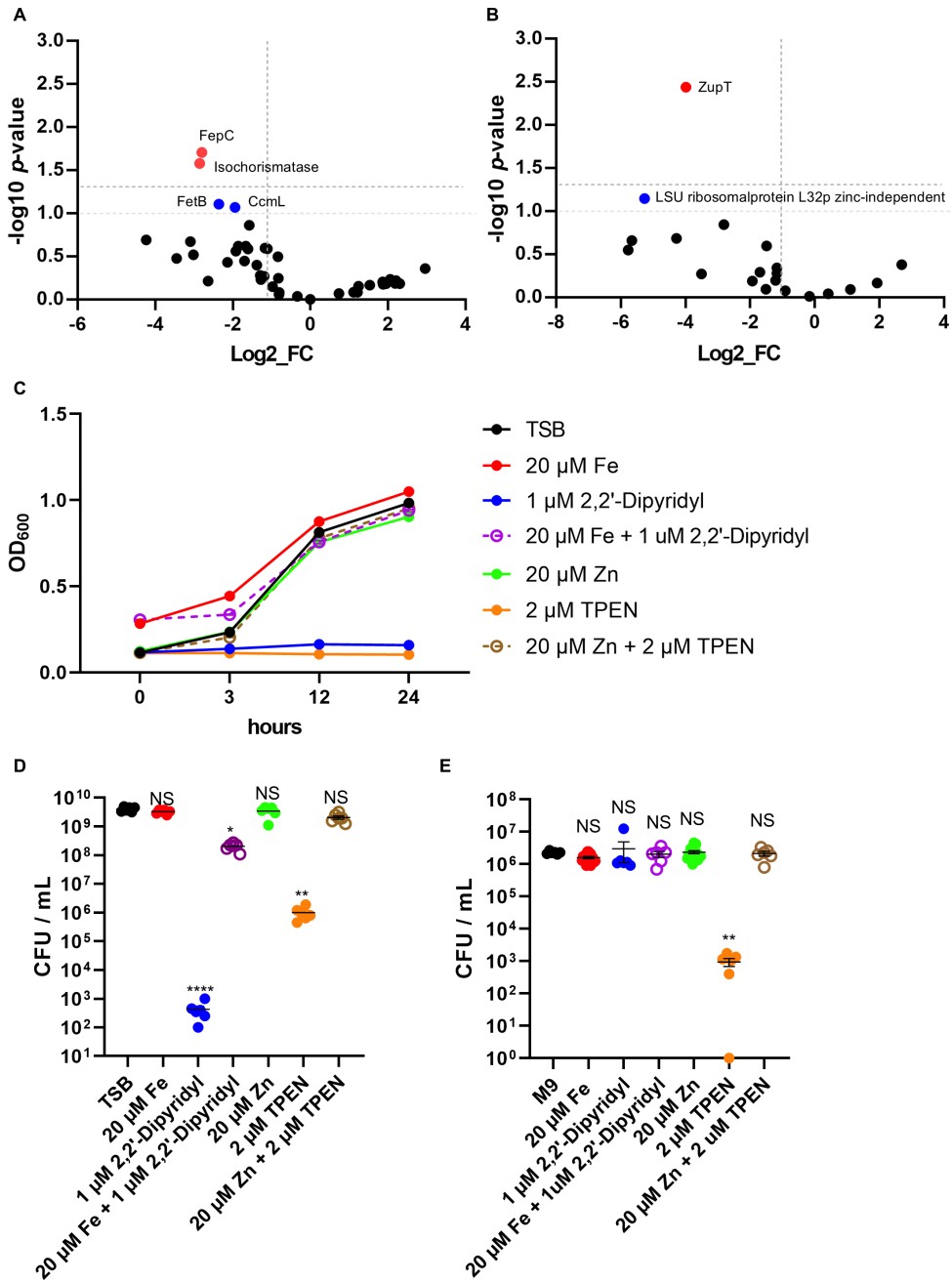

**FIG 4** Iron and zinc availability regulate *E. tarda* growth *in vitro*. (A, B) Volcano plot showing log₂ fold changes and −log₁₀ *P*-values for transposon insertion mutants recovered from *in vivo* versus *in vitro* conditions. Dotted lines indicate thresholds of log₂FC < −1 and −log₁₀ *P* > 1.3. Red dots indicate significantly depleted genes. Blue dots represent functionally related genes selected for further analysis (log₂FC < −1 and −log₁₀ *P* > 1.0). (C, D) OD600 (C) or CFU counts (D) of *E. tarda* cultured in TSB with supplementation (FeCl₃, ZnSO₄) or depletion (2,2′-dipyridyl, TPEN) of iron and zinc. (E) CFU counts of *E. tarda* grown in M9 minimal medium under the same conditions of panel D. Bars represent means ± SEM from biological replicates. Statistical analysis was performed using the Kruskal–Wallis test, followed by Dunn's multiple-comparison test with Holm–Šidák correction (each treatment vs control). *P* < 0.05, **P* < 0.01; ****P* < 0.0001; ns, not significant.

and polyamine metabolism in host environments (38, 39). Although we have not yet directly evaluated these mutants *in vivo*, the data suggest that acquisition of pyridoxal, cobalamin, and polyamines contributes to bacterial fitness during soft tissue infection, where nutrient limitation is expected. Interestingly, while thiamine (vitamin B1) is

essential for central metabolism (40), no thiamine biosynthesis genes were significantly depleted *in vivo*. A thiamine ABC transporter gene was modestly enriched but not statistically significant. This may indicate that thiamine is not limiting in necrotic tissue, or that redundant uptake pathways compensate for its acquisition.

Attempts to rescue *E. tarda* growth in minimal medium with single nutrients such as methionine, vitamin B12, or thiamine failed, indicating that auxotrophy is not driven by a single limiting factor. Rather, proliferation may depend on the combinatorial availability of essential metabolites or nutrient-rich host components, possibly liberated by tissue damage or co-infecting microbes. Only 1.22% of the transposon mutants showed significant depletion *in vivo*, clearly indicating that *E. tarda*'s_adaptation to host tissue infection involves highly specialized metabolic and nutrient acquisition pathways rather than widespread genetic determinants.

Iron is indispensable as a cofactor in enzymes involved in energy metabolism, DNA synthesis, and amino acid biosynthesis (41). *In vivo*, however, iron is tightly sequestered by host proteins, such as transferrin and lactoferrin (11, 12), creating an iron-restricted environment for pathogens. The observed attenuation of *fepC* mutants suggests that the ferric iron transporter is required to overcome this limitation (Fig. 2E and 4A; Table 4). Consistent with this, iron chelation using dipyridyl impaired *E. tarda* growth *in vitro*, especially under minimal conditions (Fig. 4E). However, exogenous iron supplementation did not enhance growth in rich medium (Fig. 4E), indicating context-dependent iron availability.

Zinc, another trace metal, is essential for numerous proteins' structural and catalytic functions (13). The broad-spectrum metal transporter ZupT was found to be critical for *in vivo* survival (Fig. 2E and 4B; Table 4). Supplementation experiments demonstrated that TPEN-mediated zinc chelation inhibited bacterial growth, whereas zinc repletion rescued it (Fig. 4D and E), confirming the physiological relevance of ZupT in zinc homeostasis and bacterial fitness. Interestingly, both zinc-dependent and zinc-independent ribosomal protein L36p variants, along with the zinc-independent L32p, were significantly depleted

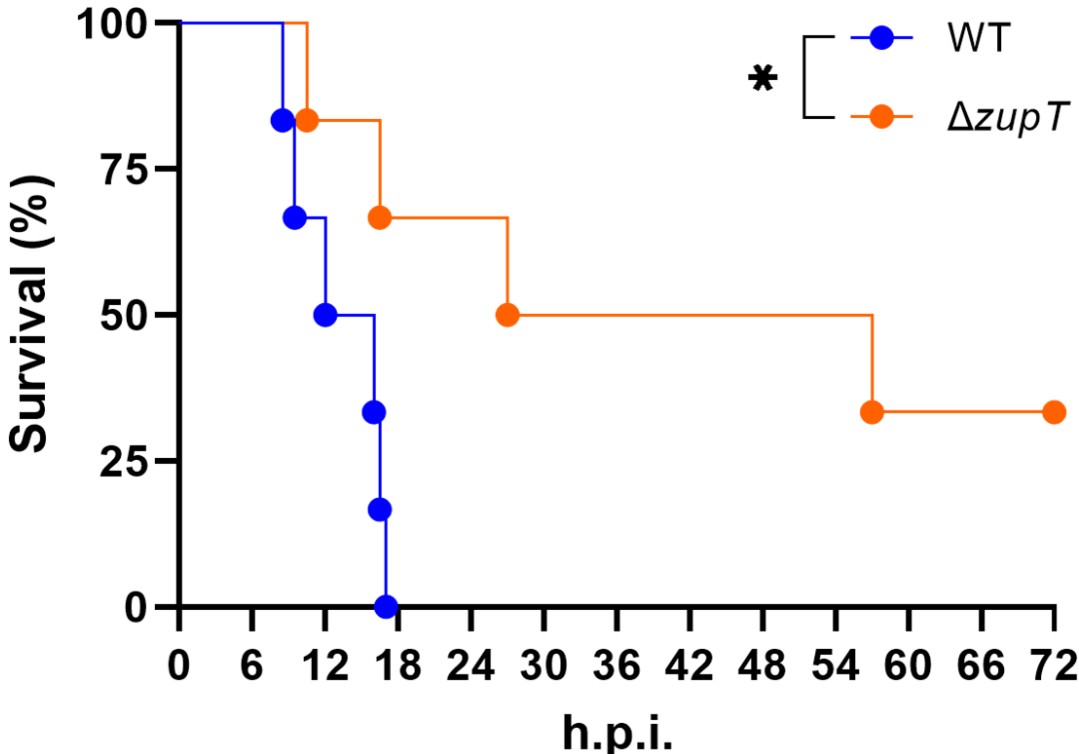

**FIG 5** *In vivo* validation of ZupT in *E. tarda* pathogenicity. Kaplan–Meier survival curves of mice inoculated subcutaneously with WT or Δ*zupT*. Statistical significance was assessed using the log-rank test. *P < 0.05.

*in vivo* (Fig. 2A). This suggests that *E. tarda* may utilize a mixed ribosomal composition—incorporating both canonical and alternative ribosomal protein isoforms—to maintain translational efficiency under fluctuating or limiting zinc conditions in host tissues. Similar ribosomal protein substitutions have been described in *Bacillus subtilis* and *Escherichia coli* during zinc limitation, supporting the idea that ribosomal remodeling is a conserved bacterial strategy to cope with host-imposed nutritional immunity (42, 43). Such ribosomal remodeling may represent a conserved bacterial strategy for sustaining protein synthesis in response to micronutrient stress. Although ZupT is indispensable for survival under systemic or late-stage zinc limitation, the Δ*zupT* mutant did not show reduced bacterial burdens in muscle tissue during early infection. This suggests that local zinc availability in the muscle is sufficient to sustain initial bacterial proliferation, and that ZupT-dependent zinc acquisition becomes critical only after the infection progresses and the host imposes stronger nutritional immunity. Consistent with these findings, the Δ*zupT* mutant exhibited significantly reduced lethality *in vivo* (Fig. 5). This provides direct experimental validation that zinc uptake is indispensable for full virulence of *E. tarda* during NSTIs.

Our findings challenge the classical toxin-centric view of pathogenesis. We also propose that metabolic fitness, particularly micronutrient acquisition, is a primary driver of *E. tarda* virulence during NSTIs. In our TraDIS screen, canonical secreted virulence factors such as toxins and cytolysins were absent among the significantly depleted mutants. However, the nature of pooled TraDIS experiments—wherein multiple transposon mutants are inoculated simultaneously—can obscure the impact of secreted or diffusible virulence factors. Such extracellular factors produced by unaffected strains may compensate functionally for mutants lacking their production, thereby masking their importance in fitness screens. Therefore, the absence of known toxins and secreted effectors in our data set does not definitively rule out their roles in pathogenesis. Additional single-mutant infection studies will be necessary to rigorously assess the contribution of secreted virulence determinants. Notably, iron sensing via the ferric uptake regulator (Fur) coordinates expression of virulence genes such as secretion systems and cytotoxins in many pathogens (41, 44, 45). In *E. tarda,* Fur-regulated pathways may link nutrient sensing to virulence activation—a hypothesis that warrants further investigation. Together, these findings highlight the importance of metabolic adaptation over canonical virulence factor expression in the context of *E. tarda*-induced NSTIs.

In summary, *E. tarda* adapts to nutrient-limited environments through a specialized set of metabolic and transport pathways. Targeting these fitness determinants, especially metal acquisition and vitamin biosynthesis, could offer novel strategies for mitigating tissue damage and halting disease progression in necrotizing infections. Our study emphasizes that metabolic fitness, rather than classical toxin-centric virulence factors, underlies the ability of *E. tarda* to cause fulminant NSTI.

## MATERIALS AND METHODS

### Bacterial strains and culture conditions

The *E. tarda* strain RIMD515001 was kindly provided by the Research Institute for Microbial Diseases, Osaka University. Bacterial strains were cultured in TSB or LaB as needed, at 25°C or 37°C with continuous shaking at 200 rpm overnight. A 1:100 dilution in fresh medium was prepared for subsequent experiments.

### Animal experiments

Five-week-old female C57BL/6 mice were purchased from Japan Jackson Laboratory.

## Bacterial preparation and infection protocol

Cultures grown at 37°C were diluted 1:100 in fresh medium and incubated for an additional 4 h. After incubation, cultures were centrifuged at 7,000 × $g$ for 3 min. The bacterial pellet was washed once with phosphate-buffered saline (PBS) and resuspended in the same medium used for culturing. Cultures grown at 25°C in LaB were centrifuged at 7,500 × $g$ for 3 min. A total of 1 × $10^7$ CFU in 100 µL of bacterial suspension was injected subcutaneously into the right thigh of each mouse. Mice were monitored daily, and those that reached the humane endpoint were euthanized.

## Bacterial burden in muscle tissue beneath the infection site

At 1, 3, and 5 days after subcutaneous injection of *E. tarda*, the infected thighs were harvested. Muscle tissue located directly beneath the injection site was dissected, homogenized in 1 mL of 0.1% gelatin in PBS using an IKA EUROSTAR homogenizer, and serially diluted 10-fold. Dilutions were plated on TSA and incubated at 37°C for 24 h. CFU was enumerated to quantify bacterial burden.

## Histological examination

At 12 h, 3 days, and 5 days post-infection, the thighs were harvested, decalcified, and embedded in paraffin. Paraffin sections (4 µm thick) containing both subcutaneous and muscle tissue were stained with hematoxylin and eosin (H&E). Each section was examined and photographed using a light microscope.

## Construction of a transposon mutant library

This study employed a transposon mutagenesis approach similar to the *V. vulnificus* model (46), with adjustments for the bacterial medium to *E. tarda*. *E. tarda* was statically cultured in lactose broth at 25°C for more than 12 h. The culture was centrifuged at 10,000 × $g$ for 1 minute, and the pellet was washed three times with lactose broth. *Escherichia coli* BW19795 carrying the plasmid pUT mini-Tn5 Km2 Tag was statically cultured in LB broth containing kanamycin at 37°C and washed in the same way. Equal volumes (500 µL) of *E. tarda* and *E. coli* suspensions were mixed and spotted onto a 1 × 1 cm piece of Hybond-C membrane (Amersham, UK) placed on LaB agar and incubated at 25°C for more than 12 h. The membrane was transferred to a tube containing 1 mL of LaB, vortexed to recover bacteria, and plated on lactose broth agar supplemented with rifampicin (100 µg/mL) and kanamycin (50 µg/mL). Colonies grown at 37°C for more than 12 h were collected as *E. tarda* transposon mutants.

## Preparation of the input pool and mouse infection

A total of 25,185 *E. tarda* transposon insertion mutants were pooled to create the input library. A 4 mL aliquot of the input pool was inoculated into 10 mL of LaB and cultured at 25°C for 16 h with shaking. The culture was diluted to OD600 = 1.0, centrifuged at 7,500 × $g$ for 3 min, and washed with 0.1% gelatin in PBS. After a second centrifugation, the pellet was resuspended in 1 mL of lactose broth. Each mouse received 100 µL of this suspension via subcutaneous injection in the right thigh. Bacterial load was confirmed by plating serial dilutions on TSA and counting CFU after incubation at 37°C for 12 h.

## Recovery of the output pool from mouse tissue

At 3 days post-infection, mice were euthanized by isoflurane inhalation. Thigh muscle at the injection site was aseptically excised, weighed, and homogenized in 1 mL of 0.1% gelatin in PBS using an IKA EUROSTAR homogenizer. Homogenates were centrifuged at 800 × $g$ for 5 min, and supernatants were plated on TSA containing rifampicin (50 µg/mL). Colonies grown at 37°C for 12 h were collected as the output pool.

## Identification of transposon insertion sites by next-generation sequencing

Genomic DNA from input and output pools was extracted using the Tissue Genomic DNA Extraction Mini Kit (FAVORGEN, Taiwan). DNA (500 ng) was fragmented to ~300 bp using a Covaris S220 ultrasonicator. Library preparation, including end repair, A-tailing, and adapter ligation, was performed using the KAPA Hyper Prep Kit. Transposon-flanking regions were amplified using a transposon-specific primer (5′-TCG TCG GCA GCG TCA GAT GTG TAT AAG AGA CAG GAT CTG ATC AAG AGA CAG-3′) and a P7 adapter primer (5′-CAA GCA GAA GAC GGC ATA CGA GAT-3′). Single-end 101 bp sequencing was performed on a HiSeq2500 (Illumina). Transposon tags were trimmed using cutadapt v1.9.1 (47), and the trimmed reads were mapped to the *E. tarda* RIMD515001 genome using Bowtie2 (v. 2.2.3) (48). Insertion sites were counted using featureCounts from the Subread package (v1.5.0-p2) (49).

## Differential insertion analysis

Raw read counts were normalized using reads per million (RPM) transformation. $Log_2$-transformed RPM values were compared between input (*in vitro* culture) and output (soft tissue infection) conditions. Differentially inserted genes were identified using a paired *t*-test and Benjamini-Hochberg correction for multiple testing (FDR < 0.05). $Log_2$ fold change ($log_2FC$) was calculated, and a volcano plot was generated to visualize genes that significantly decreased ($log_2FC < −1$, FDR < 0.05) or increased ($log_2FC > 1$, FDR < 0.05) in the infection condition.

## Growth assays under metal-limiting conditions

Bacterial growth under zinc- and iron-restricted conditions was evaluated using both minimal and rich media. Overnight cultures of wild-type and mutant strains were prepared in tryptic soy broth (TSB) at 37°C with shaking (200 rpm). Cells were harvested by centrifugation, washed once with sterile M9 minimal medium, and diluted to an initial optical density at 600 nm ($OD_{600}$) of 0.01. For minimal medium assays, cultures were inoculated into M9 supplemented with glucose (0.4%) in the presence or absence of metal supplements ($ZnSO_4$ or $FeCl_3$) at the indicated concentrations. To induce zinc limitation in rich medium, TSB was supplemented with the zinc-specific chelator N,N,N′,N′-tetrakis (2-pyridylmethyl)ethylenediamine (TPEN) (2 µM final concentration). For iron limitation, TSB was supplemented with the ferrous iron chelator 2,2′-dipyridyl (1 µM final concentration). Cultures were incubated statically or with shaking at 37°C as indicated. Optical density at 600 nm was recorded every 3 h over a 24-h period using either a spectrophotometer (tube cultures) or a plate reader (96-well format, 200 µL per well, with continuous orbital shaking). At the 24-h endpoint, viable counts were determined by serial dilution in phosphate-buffered saline and plating on LB agar to enumerate CFU. Each condition was assayed in at least three technical replicates, and the entire experiment was independently repeated on three separate occasions.

## Gene deletion of *zupT* in *E. tarda*

To generate an in-frame deletion of the *zupT* gene in *E. tarda*, we employed a two-step allelic exchange using the sacB-based suicide vector pRE112. Approximately 500-bp upstream and downstream regions flanking the *zupT* open reading frame were PCR-amplified from genomic DNA using the following primers: *zupT*_UP_fwd, 5′-cat tca tgg cca tat caa tga cgc cag cgg gaa atc cg-3′; *zupT*_UP_rev, 5′-gat gcg ctg att aca tgg gca cac tct cca aaa atg-3′; *zupT*_Down_fwd, 5′-gcccatgtaatcagcgcatcgtcctctc-3′; and *zupT*_Down_rev, 5′-gga att cat gca gtt cac ttg agt aat aac acc agc cac tat g-3′. The flanking fragments were ligated into pRE112, and the resulting deletion plasmid was transformed into *Escherichia coli* BW19795 (λpir). Conjugation into *E. tarda* was performed by filter mating. Transconjugants were selected on LB agar plates containing chloramphenicol (10 µg/mL). Single-crossover integrants were identified by chloramphenicol resistance and sucrose sensitivity. Counter-selection was performed on LB agar

supplemented with 20% sucrose to isolate double-crossover mutants. The complete deletion of *zupT* was confirmed by colony PCR using primers flanking the deleted region followed by Sanger sequencing.

## Statistical analysis

Statistical analyses were performed using GraphPad Prism version 8 (GraphPad Software, San Diego, CA). Survival curves were compared using the log-rank (Mantel–Cox) test. For multi-group comparisons, the Kruskal–Wallis test, followed by Dunn's *post hoc* test with Holm–Šidák correction (each treatment vs control), was used. Each condition included ≥3 technical replicates per experiment, and experiments were independently repeated on three separate occasions. Data are presented as means ± standard errors of the means (SEM). A *P* value of <0.05 was considered statistically significant.

## ACKNOWLEDGMENTS

This work was supported by JSPS KAKENHI Grant Numbers 19K15979 and 22K14998.

## AUTHOR AFFILIATIONS

[1]Laboratory of Veterinary Public Health, School of Veterinary Medicine, Kitasato University, Towada, Aomori, Japan

[2]Kyrgyz Research Institute of Veterinary Science named after A. Duisheev, Bishkek, Kyrgyzstan

[3]Department of Infection Metagenomics, Research Institute for Microbial Diseases, Osaka University, Suita, Osaka, Japan

[4]Department of Bacteriology I, National Institute of Infectious Diseases, Japan Institute for Health Security, Tokyo, Japan

[5]Research Institute for Microbial Diseases, The University of Osaka, Osaka, Japan

[6]Graduate School of Medicine, The University of Osaka, Osaka, Japan

## AUTHOR ORCIDs

Kohei Yamazaki  http://orcid.org/0000-0003-4862-1300
Takashige Kashimoto  http://orcid.org/0000-0002-0888-909X

## FUNDING

| Funder | Grant(s) | Author(s) |
| --- | --- | --- |
| Japan Society for the Promotion of Science | 19K15979 | Kohei Yamazaki |
| Japan Society for the Promotion of Science | 22K14998 | Kohei Yamazaki |

## AUTHOR CONTRIBUTIONS

Kohei Yamazaki, Conceptualization, Data curation, Formal analysis, Funding acquisition, Investigation, Methodology, Project administration, Resources, Validation, Visualization, Writing – original draft | Takuya Yamaguchi, Investigation | Yuichi Yokoyama, Investigation, Methodology, Supervision, Writing – review and editing | Yuka Tonosaki, Funding acquisition, Investigation, Supervision, Validation, Writing – review and editing | Klara Kursanbaeva, Investigation | Daisuke Motooka, Methodology, Software | Yukihiro Akeda, Methodology, Software, Supervision, Writing – review and editing | Takashige Kashimoto, Funding acquisition, Methodology, Project administration, Resources, Supervision, Writing – review and editing

## DATA AVAILABILITY

The whole-genome sequence of *E. tarda* strain RIMD515001 has been deposited in DDBJ under the accession numbers BAAHJJ010000001–BAAHJJ010000006.

## ETHICS APPROVAL

All animal experiments were approved by the Animal Care and Use Committee of Kitasato University (approval number 19-220) and were conducted in accordance with institutional guidelines.

## ADDITIONAL FILES

The following material is available online.

### Supplemental Material

**Fig. S1 (mSystems01657-25-s0001.tiff).** Histopathological changes in soft tissue following infection with *E. tarda* cultured in LaB at 25°C.
**Fig. S2 (mSystems01657-25-s0002.tiff).** Validation of random transposon insertion by Southern blotting.

### Open Peer Review

**PEER REVIEW HISTORY (review-history.pdf).** An accounting of the reviewer comments and feedback.

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
