## [Reviewer comments · mSystems]

Nutrient Acquisition Drives *Edwardsiella tarda* Pathogenesis in Necrotizing Soft Tissue Infection

Kohei Yamazaki, Takuya Yamaguchi, Yuichi Yokoyama, Yuka Tonosaki, Klara Kursanbaeva, Daisuke Motooka, Yukihiro Akeda, and Takashige KASHIMOTO

Corresponding Author(s): Takashige KASHIMOTO, Kitasato Daigaku

Review Timeline:

Submission Date:

November 25, 2025

Accepted:

December 17, 2025

Editor: Sophie Darch

Reviewer(s): Disclosure of reviewer identity is with reference to reviewer comments included in decision letter(s). The following individuals involved in review of your submission have agreed to reveal their identity: Keita Nishiyama (Reviewer #1)

Transaction Report:

DOI: <https://doi.org/10.1128/msystems.01657-25>

Re: mSystems01657-25 (**Nutrient Acquisition Drives *Edwardsiella tarda* Pathogenesis in Necrotizing Soft Tissue Infection**)

Dear Prof. Takashige KASHIMOTO:

Your manuscript has been accepted, and I am forwarding it to the ASM production staff for publication. Your paper will first be checked to make sure all elements meet the technical requirements. ASM staff will contact you if anything needs to be revised before copyediting and production can begin. Otherwise, you will be notified when your proofs are ready to be viewed.

Sincerely,
Sophie Darch
Editor
mSystems